# Multiple sensors provide spatiotemporal oxygen regulation of gene expression in a *Rhizobium*-legume symbiosis

**Paul J. Rutten**[1], **Harrison Steel**[2], **Graham A. Hood**[3], **Vinoy K. Ramachandran**[1], **Lucie McMurtry**[1], **Barney Geddes**[1], **Antonis Papachristodoulou**[2], **Philip S. Poole**[1]*

**1** Department of Plant Sciences, University of Oxford, Oxford, United Kingdom, **2** Department of Engineering Science, University of Oxford, Oxford, United Kingdom, **3** Department of Molecular Microbiology, John Innes Centre, Norwich, United Kingdom

* philip.poole@plants.ox.ac.uk

**Data Availability Statement:** All relevant data are within the manuscript and its Supporting Information files.

## Abstract

Regulation by oxygen ($O_2$) in rhizobia is essential for their symbioses with plants and involves multiple $O_2$ sensing proteins. Three sensors exist in the pea microsymbiont *Rhizobium leguminosarum* Rlv3841: hFixL, FnrN and NifA. At low $O_2$ concentrations (1%) hFixL signals via FxkR to induce expression of the FixK transcription factor, which activates transcription of downstream genes. These include *fixNOQP*, encoding the high-affinity $cbb_3$-type terminal oxidase used in symbiosis. In free-living Rlv3841, the hFixL-FxkR-FixK pathway was active at 1% $O_2$, and confocal microscopy showed hFixL-FxkR-FixK activity in the earliest stages of Rlv3841 differentiation in nodules (zones I and II). Work on Rlv3841 inside and outside nodules showed that the hFixL-FxkR-FixK pathway also induces transcription of *fnrN* at 1% $O_2$ and in the earliest stages of Rlv3841 differentiation in nodules. We confirmed past findings suggesting a role for FnrN in *fixNOQP* expression. However, unlike hFixL-FxkR-FixK, Rlv3841 FnrN was only active in the near-anaerobic zones III and IV of pea nodules. Quantification of *fixNOQP* expression in nodules showed this was driven primarily by FnrN, with minimal direct hFixL-FxkR-FixK induction. Thus, FnrN is key for full symbiotic expression of *fixNOQP*. Without FnrN, nitrogen fixation was reduced by 85% in Rlv3841, while eliminating hFixL only reduced fixation by 25%. The hFixL-FxkR-FixK pathway effectively primes the $O_2$ response by increasing *fnrN* expression in early differentiation (zones I-II). In zone III of mature nodules, near-anaerobic conditions activate FnrN, which induces *fixNOQP* transcription to the level required for wild-type nitrogen fixation activity. Modelling and transcriptional analysis indicates that the different $O_2$ sensitivities of hFixL and FnrN lead to a nuanced spatiotemporal pattern of gene regulation in different nodule zones in response to changing $O_2$ concentration. Multi-sensor $O_2$ regulation is prevalent in rhizobia, suggesting the fine-tuned control this enables is common and maximizes the effectiveness of the symbioses.

**Funding:** This work was supported by the Biotechnology and Biological Sciences Research Council [grant numbers BB/L011484/1 and BB/M011224/1]. AP and HS were supported by the Engineering and Physical Sciences Research Council [grant number EP/M002454/1]. The funders played no role in the study design, data collection, decision to publish or preparation of the manuscript. https://bbsrc.ukri.org.

**Competing interests:** No competing interests.

## Author summary

Rhizobia are soil bacteria that form a symbiosis with legume plants. In exchange for shelter from the plant, rhizobia provide nitrogen fertilizer, produced by nitrogen fixation. Fixation is catalysed by the nitrogenase enzyme, which is inactivated by oxygen. To prevent this, plants house rhizobia in root nodules, which create a low oxygen environment. However, rhizobia need oxygen, and must adapt to survive the low oxygen concentration in the nodule. Key to this is regulating their genes based on oxygen concentration. We studied one *Rhizobium* species which uses three different protein sensors of oxygen, each turning on at a different oxygen concentration. As the bacteria get deeper inside the plant nodule and the oxygen concentration drops, each sensor switches on in turn. Our results also show that the first sensor to turn on, hFixL, primes the second sensor, FnrN. This prepares the rhizobia for the core region of the nodule where oxygen concentration is lowest and most nitrogen fixation takes place. If both sensors are removed, the bacteria cannot fix nitrogen. Many rhizobia have several oxygen sensing proteins, so using multiple sensors is likely a common strategy enabling rhizobia to adapt to low oxygen precisely and in stages during symbiosis.

## Introduction

Rhizobia are alpha-proteobacteria that engage in symbiosis with legume plants [1]. The bacteria convert inert atmospheric $N_2$ into biologically accessible ammonia and provide it to their plant host in a process called nitrogen fixation [2,3]. All biological fixation is catalysed by the nitrogenase enzyme complex that evolved before the Great Oxygenation Event and requires near-anoxic conditions to function [4–6]. However, rhizobia are obligate aerobes and must respire to meet the high energy demands of nitrogen fixation [7,8]. These competing requirements create an 'oxygen paradox' in symbiotic nitrogen fixation [9,10]. To overcome this paradox, intricate cooperation between rhizobia and their plant partners has evolved (reviewed in [11,12]). Legume plants host rhizobia in dedicated root nodules which form where bacteria have entered the plant root, usually via infection threads (reviewed in [13,14]). Nodules create a near-anoxic internal environment suitable for nitrogenase activity [15–17]. To produce this environment, oxygen ($O_2$) is captured and shuttled to bacteroids by plant leghaemoglobins [18–21]. The concentration of remaining free $O_2$ in the core nitrogen fixation zone of nodules is as low as 20–50 nM [22,23]. Rhizobia undergo a radical lifestyle change after nodule entry to survive and fix nitrogen in these conditions (reviewed in [24,25]). In indeterminate nodules, such as those produced by *Pisum sativum* (pea), rhizobia are initially free-living upon entry [26,27]. They then undergo irreversible lifestyle changes as they move from the nodule tip to its core [28,29]. Beginning in zone II and accelerating in the II-III interzone, rhizobia terminally differentiate into quasi-organelle bacteroids specialized for nitrogen fixation [30,31]. Zone III of indeterminate nodules contains differentiated bacteroids which are actively fixing nitrogen [32]. Rhizobial regulatory mechanisms sensitive to $O_2$ tension are essential for successful differentiation into bacteroids and the establishment of a productive symbiosis [33–35].

Multiple $O_2$ sensors have evolved in rhizobia, three of which are widespread and often coexist within the same organism [11,36]. The first is the membrane-bound FixL protein, which forms a two-component system (TCS) with the FixJ receiver protein (reviewed in [37,38]). Under microaerobic conditions, FixL phosphorylates FixJ, which in turn induces expression of the *fixK* transcription factor [39–41]. FixK induces expression of downstream genes by

binding as a dimer to an 'anaerobox' motif (TTGAT-$N_4$-ATCAA) upstream of their promoters [42,43].

The second common $O_2$ sensor is a variant of FixL called hybrid FixL (hFixL) [44,45]. This forms an alternative TCS with FxkR acting as the receiver protein. FxkR is not a FixJ homolog but similarly induces expression of *fixK*, by binding to an upstream 'K-box' motif (GTTACA-$N_4$-GTTACA) [46]. The third $O_2$ sensor is the FnrN transcription factor. Like FixK, FnrN binds the anaerobox motif as a dimer and both are close homologs of the *E. coli* anaerobiosis regulator FNR [47–49]. Unlike FixK but like FNR, FnrN contains an N-terminal cysteine-rich cluster that makes the protein a direct sensor of $O_2$ [50–53]. The FixL and hFixL sensors are known to become active at relatively mildly microaerobic conditions, including in free-living rhizobia [54–56]. FnrN is likely to be far less $O_2$ tolerant. The $O_2$ sensitivity of FnrN has not been determined, but the *E. coli* FNR homolog is active only under anaerobic conditions [57–59]. All symbiotic rhizobia studied to date employ at least one of these three sensors [11]. It is common for these sensors to coexist, notably in *Rhizobium leguminosarum* biovar *viciae* VF39, multiple strains of *Ensifer meliloti* (previously *Sinorhizobium meliloti*) and *Rhizobium etli* CFN42 [44,45,60–62].

Further emphasizing the importance of $O_2$ regulation in symbiotic nitrogen fixation, rhizobia also employ the $O_2$ sensing NifA transcription factor to regulate their final differentiation into nitrogen fixing bacteroids (for reviews see [38,63]). NifA oxygen sensitivity is thought to derive from a metal-binding cysteine-rich motif in an inter-domain linker of the protein [64–66]. The protein has a large regulon, notably including nitrogenase components such as *nifH* [67–70]. Expression of *nifA* is typically auto-regulated in rhizobia, often via read-through from an upstream gene or operon that is NifA regulated, in many cases *fixABCX* [69,71–73]. In Rlv3841, a *fixABCX* operon is found directly upstream of *nifA*, suggesting such a read-through NifA auto-activation mechanism. Usually, neither expression of *nifA* nor the activity of the protein is directly regulated by the three $O_2$ sensors described above [11]. One notable exception is *E. meliloti*, where *nifA* is regulated by the FixLJ system [74–76]. There is no evidence that FixK or FnrN directly regulates *nifA* expression in Rlv3841.

There appears to be a spectrum among rhizobia, with some species segregating oxygen sensors into separate pathways, whilst in other species these sensors have partially or completely merged into a combined hierarchical pathway [77,78]. Where oxygen sensors are in separated pathways, redundancy often exists. In these situations the loss of one oxygen sensor does not abolish nitrogen fixation activity [79,80]. By contrast, where sensors have been merged into a single regulatory pathway, some components are individually essential [11]. Thus, loss of one oxygen sensor can severely impair nitrogen fixation even if other sensors remain.

In *R. leguminosarum* bv. VF39, knocking out FnrN or hFixL reduced nitrogen fixation to 30% or 50% of WT respectively, suggesting a non-hierarchical, redundant arrangement [60]. The hFixL-FxkR-FixK pathway of *R. etli* CFN42 is dispensable as a double *fixK* mutant had no effect on nitrogen fixation, whilst a double *fnrN* mutant reduced fixation to 20% of WT levels. By contrast, *R. etli* CFN42 appears to employ a complex hierarchical pathway, with multiple homologs of FixK and FnrN regulating each other's expression [61,62]. Species encoding homologs of only hFixL or FnrN have also been found. *Rhizobium leguminosarum* biovar *viciae* UPM791 contains two FnrN homologs but neither FixL nor hFixL [81]. It is unknown whether the two FnrN proteins respond to different $O_2$ concentrations or act in a redundant fashion. *E. meliloti* 1021 contains no FnrN homolog but a well-studied FixLJ system and appears to have homologs of hFixL and FxkR [82–84].

To examine the relationship between hFixL and FnrN, we studied the model organism *Rhizobium leguminosarum* biovar *viciae* 3841 (Rlv3841) which employs both sensors (Fig 1) [85,86]. Rlv3841 has a single chromosome whose gene names start with RL, and six

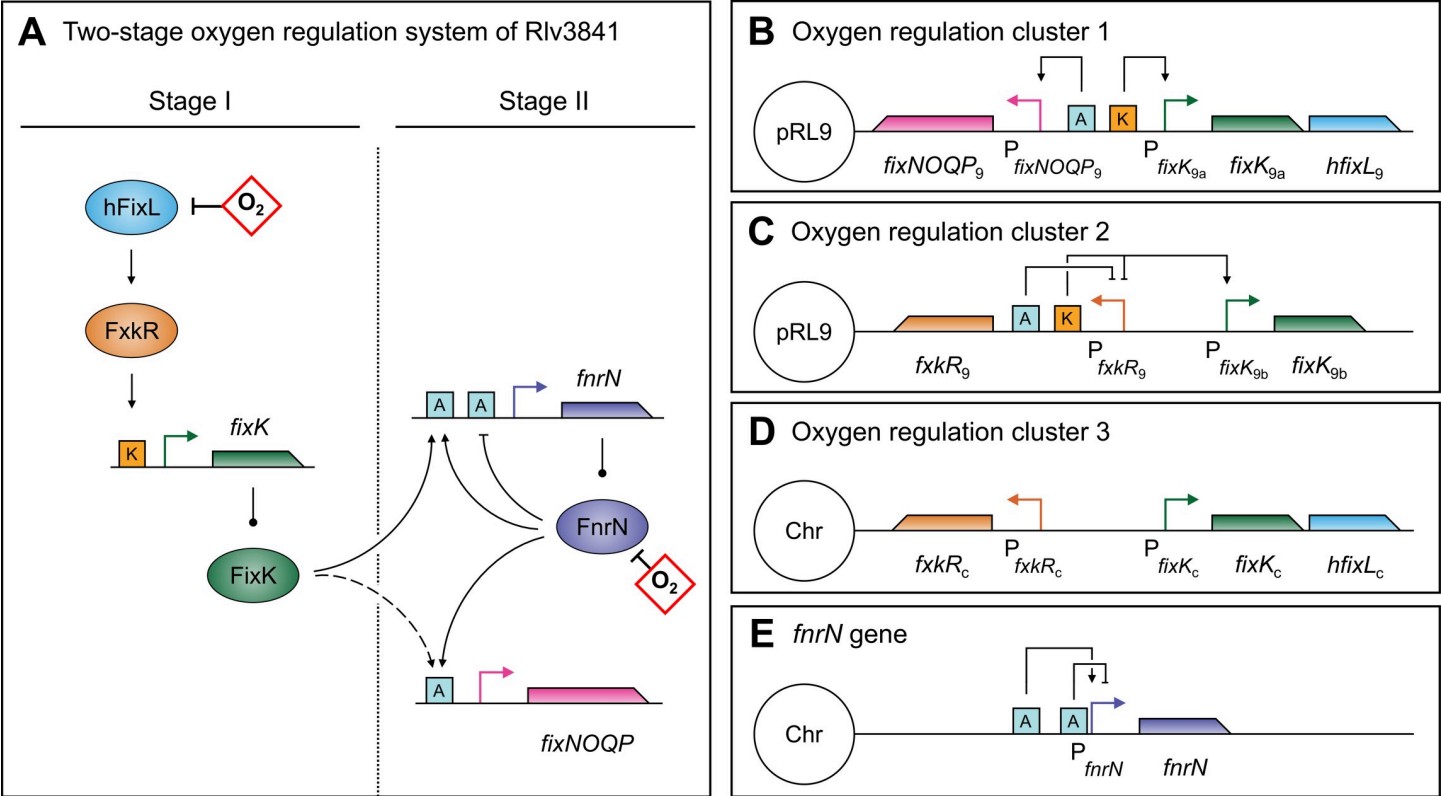

**Fig 1. The integrated hFixL-FxkR-FixK and FnrN oxygen regulation systems of Rlv3841 form a single pathway and are genetically clustered.** Oxygen is shown in red diamonds. Proteins are shown as ovals, operator sites as squares and genes as pointed rectangles. Transcription start sites are shown as right-angled arrows. Line endings indicate activation (arrows), inhibition (blunt end) and translation (circle). (**A**) The single pathway formed by the two sensors acts in two stages. Stage I starts under microaerobic conditions and can function outside the nodule. In this stage, hFixL is active but FnrN is not. hFixL activates FxkR, which binds to the K-box operator (orange "K" squares) to induce expression of *fixK*. FixK binds to anaerobox operators (blue "A" squares) to induce expression, including upstream of *fixNOQP* (dashed line) and *fnrN*. Once oxygen in the bacteria reaches near-anaerobic levels, FnrN becomes active and stage II begins. Like FixK, FnrN binds anaeroboxes. It auto-regulates *fnrN* both positively and negatively and induces *fixNOQP* expression. (**B**) Rlv3841 has multiple copies of several oxygen regulation genes and many are arranged in clusters. On megaplasmid pRL9, *fixK9a* forms an operon with *hfixL9*, regulated by a K-box. This operon is adjacent to *fixNOQP9*, regulated by an anaerobox. (**C**) *fixK9b* and *fxkR9* are adjacent, with an anaerobox and a K-box in their intergenic region. (**D**) The Rlv3841 chromosome also has a cluster, containing *fxkRc*, *fixKc* and *hfixLc*. Unlike the similar clusters on pRL9, the intergenic region of this cluster contains no anaerobox or K-box operators. (**E**) The *fnrN* gene is not part of a cluster and is positively and negatively regulated by a distal and proximal anaerobox, respectively. Details of transcription start site, anaerobox and K-box locations can be found in Table 1.

megaplasmids pRL7-12 whose gene names start with e.g. pRL9. The main symbiotic plasmid is pRL10, but many symbiotic genes are also found on pRL9 including a copy of the *fixNOQP* and *fixGHIS* operon. Rlv3841 encodes two copies of *hfixL*, which we named *hfixL9* (pRL90020 on pRL9) and *hfixLc* (RL1879 on the chromosome), with 54.9% identity at the protein level. The strain also contains two homologs (58% identity) of *fxkR*, *fxkR9* (pRL90026) and *fxkRc* (RL1881). It has three putative *fixK* genes, which we designated *fixK9a* (pRL90019), *fixK9b* (pRL90025) and *fixKc* (RL1880). The *fixK9a* and *fixK9b* sequences have 53% amino acid identity, whilst *fixKc* shares 38% and 47% identity with these proteins, respectively. Both *fixK9a*-*hfixL9* and *fixKc*-*hfixLc* appear to form operons (Fig 1B and 1D). Rlv3841 has one copy of *fnrN* (RL2818), regulated by two anaeroboxes. A similar dual-anaerobox arrangement exists in Rlv UPM791, where FnrN positively and negatively auto-regulates its own expression [87]. Binding of FnrN to the distal anaerobox induces *fnrN* transcription and binding to the proximal anaerobox represses it. Auto-activation of FnrN has also been reported in *Rhizobium etli* CNPAF512 [88]. FixK regulation of *fnrN* expression is likely as it also binds anaeroboxes, but this had not been investigated.

A study in *R. leguminosarum* VF39 found that microaerobic expression of *fnrN* also requires RpoN [89]. This finding has not been replicated elsewhere, and its significance remains unclear. Work in *R. etli* CNPAF512 showed that *fnrN* is not controlled by RpoN in that organism [88]. Rlv3841 encodes one putative *rpoN* gene (RL0422), but we found no RpoN binding sites upstream of the Rlv3841 *fnrN* transcription start site. RpoN therefore does not appear to be required for *fnrN* expression in Rlv3841.

A parallel arrangement of hFixL-FxkR-FixK and FnrN in Rlv3841 would produce redundancy, whereas an arrangement in series would create hierarchy between the two regulators. Our goal is therefore to understand how the two sensors interact in Rlv3841 and to provide insight into why they coexist.

## Results

### Expression of *fnrN* is auto-regulated and controlled by the hFixL-FxkR-FixK pathway

The hFixL-FxkR-FixK pathway is known to be active at relatively high $O_2$ concentrations, including in free-living rhizobia under microaerobic conditions [45,90]. The role of FnrN is less well understood and we began by investigating this sensor. The *fnrN* gene contains two anaeroboxes upstream of its promoter, a proximal site at -2 relative to the transcription start site (TSS) and a distal site at -34 (Fig 1E and Table 1). Binding at the proximal site inhibits transcription through steric hindrance, whilst the distal site activates it [87]. As expected from the presence of the distal site, *fnrN* was induced under microaerobic conditions in free-living Rlv3841 (Fig 2). Both FixK and FnrN bind to anaerobox operators and induce gene expression under microaerobic conditions [91,92]. Therefore, microaerobic induction of *fnrN* could be due to FnrN auto-activation and/or induction by the hFixL-FxkR-FixK pathway. To determine their respective importance, expression of *fnrN* was studied in Rlv3841 mutants defective in either hFixL or FnrN.

In a double *hfixL* mutant (LMB496; *hfixL$_9$*::ΩSpec *hfixL$_c$*:pK19 single recombination), free-living microaerobic expression of *fnrN* was reduced to 25% of its wild-type (WT) level (Fig 3).

**Table 1. Location of transcription start sites, anaeroboxes and K-boxes for select oxygen regulation genes in Rlv3841.**

| Gene name | TSS coordinate | Anaerobox 1 location | Anaerobox 2 location | K-box location |
|---|---|---|---|---|
| *fnrN* (RL2818) | 2978390 | -34 | -2 | Not present |
| *fixK$_{9a}$* (pRL90019) | 19878 | Not present | Not present | -62 |
| *fixK$_{9b}$* (pRL90025) | 27090 | Not present | Not present | -62 |
| *fixK$_c$* (RL1880) | 1977111 | Not present | Not present | Not present |
| *fxkR$_9$* (pRL90026) | 27146 | Not present | +38 | +6 |
| *fxkR$_c$* (RL1881) | 1977167 | Not present | Not present | Not present |
| *fixNOQP$_9$* (pRL90018-16) | 19721 | -33 | Not present | Not present |
| *fixNOQP$_{10}$* (pRL100205-207) | 206214 | -34 | Not present | Not present |
| *fixGHIS$_9$* (pRL90015-12A) | 15908 | -32 | Not present | Not present |
| *fixGHIS$_{10}$* (pRL100208) | 210004 | -32 | Not present | Not present |

Locations of anaeroboxes and K-boxes are given relative to the TSS of each gene respectively. Anaeroboxes more than 90 bp upstream of transcription start sites are not included. The location of activating anaeroboxes is well conserved across the genes shown above (between -32 and -34 relative to the TSS). Likewise, both megaplasmid-encoded *fixK* copies have their respective K-boxes in the same position relative to the TSS (-62). The anaerobox downstream (+38) of the *fxkR$_9$* transcription start site likely represses it when bound. A single K-box is shared between *fxkR$_9$* and *fixK$_{9b}$* (see Fig 1). It is correctly located (-62) to induce *fixK$_{9b}$* when bound, and its location suggests it simultaneously represses *fxkR$_9$* (+6). The operators downstream of the *fxkR$_9$* TSS may create a negative feedback loop in the Rlv3841 hFixL-FxkR-FixK pathway. The downstream anaerobox also suggests FnrN repression of hFixL-FxkR-FixK via repression of *fxkR$_9$* transcription, but this requires further study.

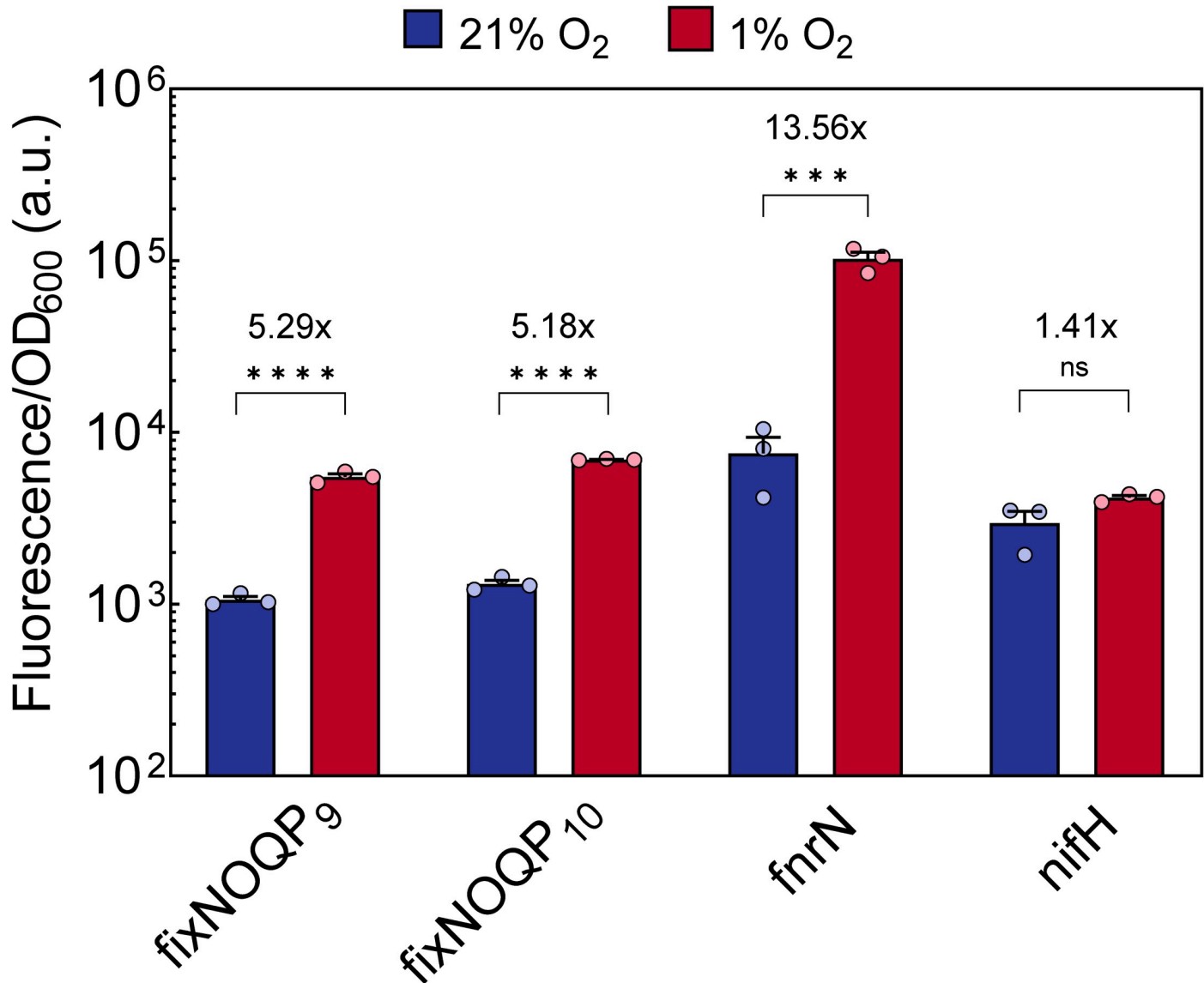

**Fig 2. Microaerobiosis induces *fixNOQP* and *fnrN* genes in free-living Rlv3841.** Promoter fusions of *fixNOQP*$_9$ (OPS1267), *fixNOQP*$_{10}$ (OPS1287) and *fnrN* (OPS1296) were used to measure the activity of these promoters in free-living Rlv3841 at 1% $O_2$ (red bars) relative to 21% $O_2$ (blue bars). Activity of all three promoters, measured as fluorescence normalised by $OD_{600}$, increased under microaerobic conditions. A positive control (OPS1294) showed no impact on OD-normalised fluorescence due to the microaerobic environment. A similar fold induction of ~5 was recorded for both *fixNOQP* operons, but *fnrN* showed more than double this fold change indicating stronger induction. No effect of $O_2$ concentration on *nifH* expression (OPS1268) was observed, indicating no NifA activity. Values are plotted on a logarithmic scale. Data are averages (±SEM) from three biological replicates, ns (not significant) P ≥ 0.05; \*\*\*P < 0.001; \*\*\*\*P < 0.0001; by Student's t test.

The single mutant of *hfixL*$_9$ (LMB495; *hfixL*$_9$::ΩSpec) individually reproduced most of this reduction whilst the single mutant of *hfixL*$_c$ (LMB403; *hfixL*$_c$:pK19 single recombination) did not reduce *fnrN* expression. This suggests hFixL$_9$ is the critical hFixL protein under free-living microaerobic conditions, with hFixL$_c$ playing little to no role.

hFixL acts through the FxkR intermediary in rhizobia and Rlv3841 contains two FxkR homologs [44]. *fxkR*$_9$ forms an $O_2$ regulation cluster with *fixK*$_{9b}$ which we labelled cluster 2 (Fig 1C) and *fxkR*$_c$ forms a cluster with *fixK*$_c$-*hfixL*$_c$, which we labelled cluster 3 (Fig 1D). Cluster 2 contains an anaerobox and a K-box. The anaerobox is relatively far from the *fixK*$_{9b}$ TSS

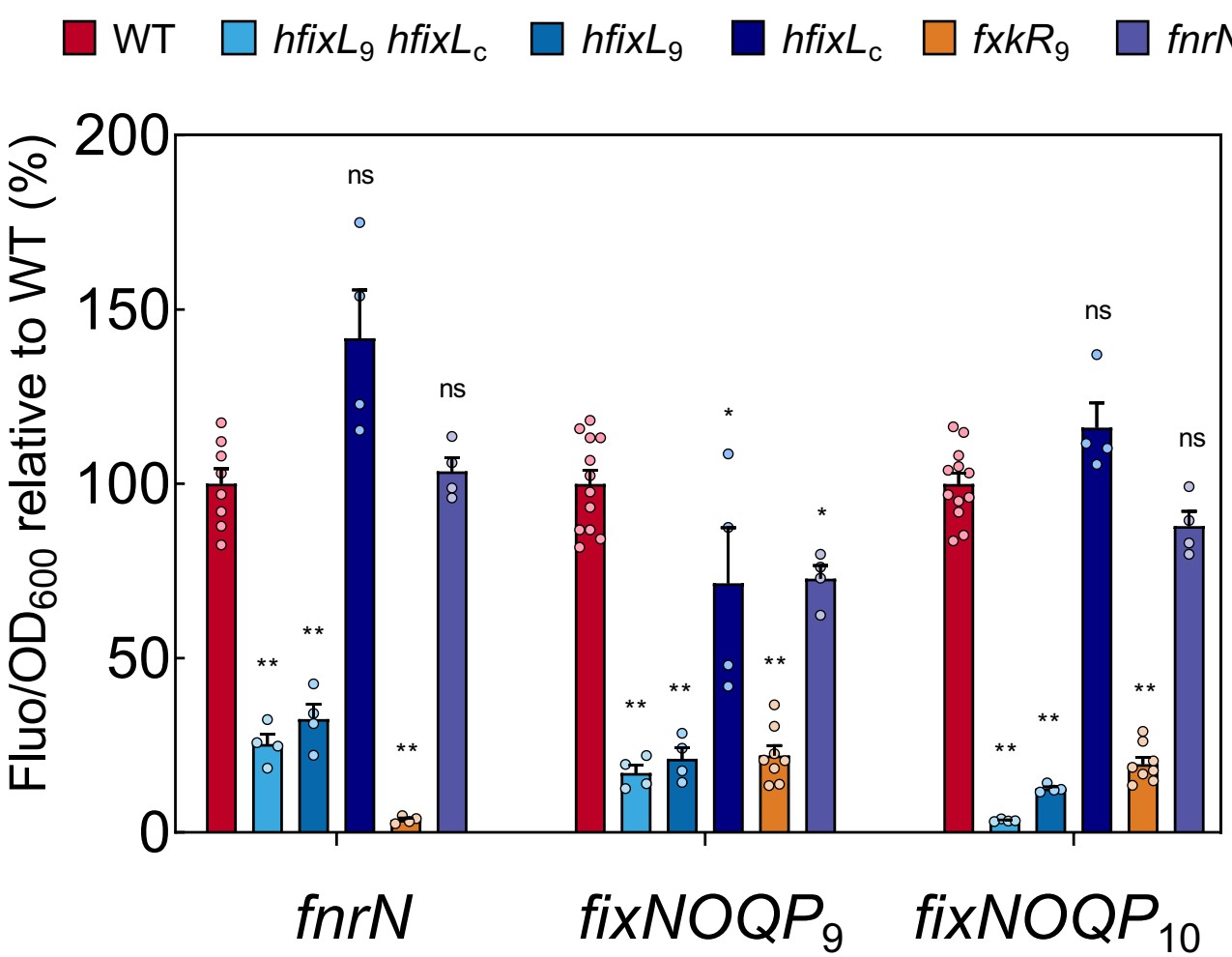

**Fig 3. Under microaerobic (1% O$_2$) conditions in free living cells, the hFixL-FxkR-FixK pathway and not FnrN is a key activator of anaerobox controlled genes.** Plasmids with promoter fusions to *syfp2* for *fnrN* (pOPS0980), *fixNOQP$_9$* (pOPS0978) and *fixNOQP$_{10}$* (pOPS0977) were conjugated into Rlv3841 WT and O$_2$ regulation mutants (*hfixL$_9$ hfixL$_c$*, LMB496; *hfixL$_9$*, LMB495; *hfixL$_c$*, LMB403; *fxkR$_9$*, OPS1808; *fnrN*, LMB648). Fluorescence and OD$_{600}$ measurements were taken after cells were grown under microaerobic conditions (1% O$_2$). Individual values (Fluo/OD$_{600}$) are normalised such that the WT average is 100% for each reporter group. Activity from all three promoters was critically reduced or nearly abolished in the double *hfixL* and *fxkR* mutant backgrounds. The *hfixL$_9$* homolog had a far more pronounced effect on expression of all three genes than did the *hfixL$_c$* homolog. Little or no reduction in expression was observed when *fnrN* was mutated. Data are averages (±SEM) from at least four biological replicates. Statistical tests are differences relative to WT expression; ns (no significant decrease) P ≥ 0.05; *P < 0.001; **P < 0.0001; by one-way ANOVA with Dunnett's post-hoc test for multiple comparisons.

(-94), but downstream of the *fxkR$_9$* TSS, suggesting its main role is to repress *fxkR$_9$* transcription (Table 1). Its effect on *fixK$_{9b}$* transcription may be minimal, if any. The K-box is also downstream of the *fxkR$_9$* TSS, suggesting it also represses transcription. Simultaneously, this K-box also likely acts to induce *fixK$_{9b}$*, as its location upstream of the *fixK$_{9b}$* TSS (-62) is identical to the relative position of the upstream *fixK$_{9a}$* K-box in cluster 1 (Fig 1B and Table 1). Both K-boxes therefore appear functional for *fixK* induction, and the cluster 2 K-box appears to have a dual function, repressing *fxkR$_9$* and inducing *fixK$_{9b}$*. The second cluster, containing *fxkR$_c$*, has no anaeroboxes or K-boxes (Fig 1D). This could imply minimal expression from the genes in this cluster, or constitutive expression that is not O$_2$ regulated. Neither *fixK$_c$* (1.6-fold upregulated, p = 0.140) nor *fxkR$_c$* (1.3-fold upregulated, p = 0.145) was significantly upregulated in 21 day old bacteroids compared to free-living Rlv3841 [93]. Based on these findings,

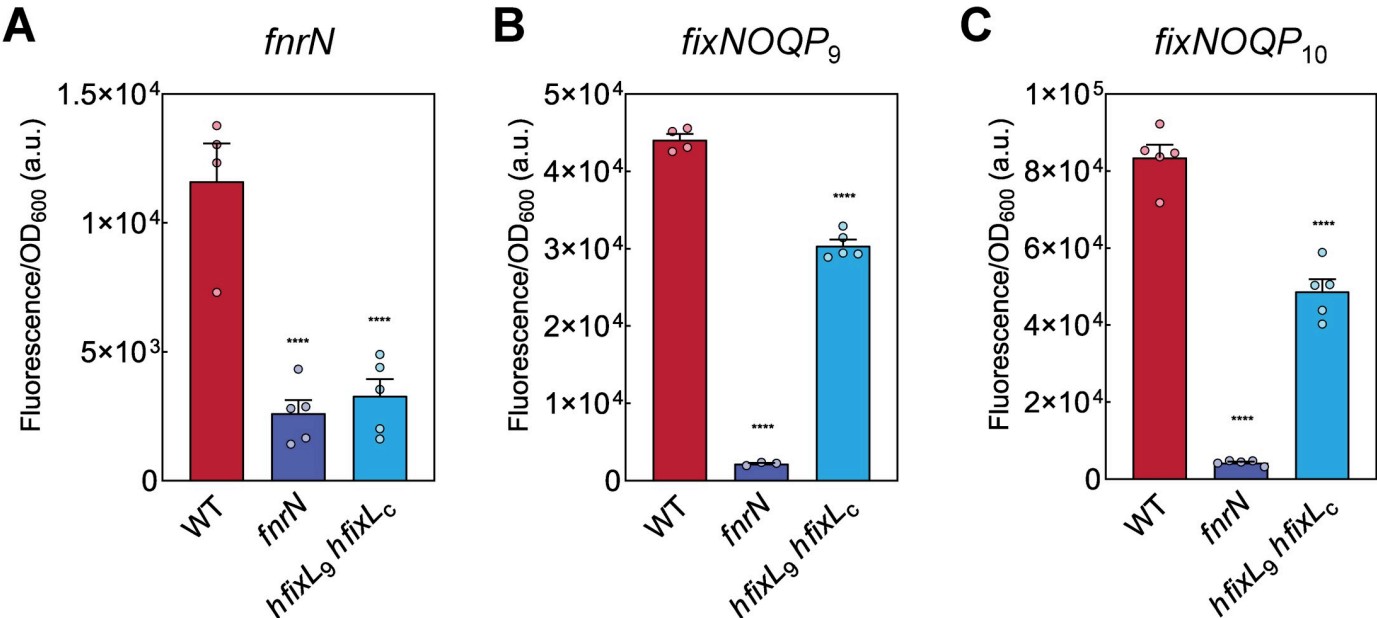

**Fig 4. *In planta*, *fnrN* is both auto-regulated and controlled by the hFixL-FxkR-FixK pathway, whilst the *fixNOQP* operons are primarily controlled by FnrN.** Rlv3841 WT and mutant strains (*fnrN*, LMB648; *hfixL$_9$ hfixL$_c$*, LMB496) containing promoter fusions to *syfp2* for *fnrN* (pOPS0980), *fixNOQP$_9$* (pOPS0978) and *fixNOQP$_{10}$* (pOPS0977) were inoculated on plants and bacteroids isolated for measurements. Expression in bacteroids of *fnrN* (**A**) is impaired in the *fnrN* background where auto-activation cannot take place. Expression of *fnrN* is similarly impaired in the double *hfixL* mutant, indicating the hFixL-FxkR-FixK pathway also plays an important role in symbiotic *fnrN* induction. Expression of *fixNOQP$_9$* (**B**) and *fixNOQP$_{10}$* (**C**) is significantly reduced in the double *hfixL* mutant and almost abolished in the *fnrN* mutant. Thus both FnrN and the hFixL-FxkR-FixK pathway play an important role in the expression of all three genes. Data are averages (±SEM) from at least three plants, $^{****}P < 0.0001$; by one-way ANOVA with Dunnett's post-hoc test for multiple comparisons.

we speculated that FxkR$_9$ is the main FxkR protein and *fxkR$_9$* was deleted to produce strain OPS1808 (Δ*fxkR$_9$*). This mutant reproduced the reduced induction of *fnrN* under free-living microaerobic conditions observed in the double *hfixL* mutant (Fig 3). This finding supports the role of FxkR$_9$ as the mediator of hFixL O$_2$ regulation in Rlv3841, in agreement with studies in other rhizobia [44,45]. Studying the role of the hFixL-FxkR-FixK pathway in *fnrN* expression *in planta*, we observed that the double *hfixL* mutant reduced *fnrN* expression to 28% of WT levels (Fig 4A). This indicates the pathway also plays an important role in inducing *fnrN* during symbiosis.

We next studied the role of FnrN auto-regulation. A mutant of *fnrN* (LMB648, *fnrN*::ΩTet) had no effect on expression of *fnrN* at 1% O$_2$ (Fig 3), indicating FnrN auto-activation does not occur under free-living microaerobic conditions. By contrast, *in planta* the *fnrN* mutant reduced *fnrN* expression to 22% of WT levels (Fig 4A), similar to the reduction observed in the *hfixL* double mutant. FnrN auto-activation is thus an important regulatory effect during symbiosis but not under microaerobic conditions. During symbiosis, expression of *fnrN* is driven both by auto-activation and the hFixL-FxkR-FixK pathway, and both are required to attain full WT-level expression of the gene.

## FnrN is critical for symbiotic gene expression but hFixL also plays an important role

To understand the respective importance of FnrN and hFixL as regulators of anaerobox controlled genes during symbiosis, their role in *fixNOQP* expression was studied. The *fixNOQP* operon encodes a high-affinity *cbb$_3$*-type terminal oxidase required for respiration during symbiosis [94–96]. It is typically regulated by an anaerobox [97]. Some rhizobia

encode multiple redundant terminal oxidases controlled by different regulators, but no alternatives appear to be encoded by Rlv3841 [98,99]. The strain therefore likely relies entirely on *fixNOQP* for respiration during symbiosis. Three putative homologs of *fixNOQP* exist in Rlv3841, which we labelled *fixNOQP*$_9$ (encoded on pRL9), *fixNOQP*$_{10}$ (encoded on pRL10) and *fixNOQP*$_c$ (encoded on the chromosome) [85]. Rlv VF39 encodes two *fixNOQP* operons, and either was able to sustain nitrogen fixation activity [56]. In Rlv3841, the plasmid-encoded operons are near-identical (>90% protein identity) but diverge from the *fixNOQP*$_c$ operon, with which they share approximately 50% identity. Only the plasmid-encoded *fixNOQP*$_9$ and *fixNOQP*$_{10}$ operons contain an upstream anaerobox, at -33 and -34 relative to their TSS respectively (Table 1). Past microarray work in our group found no significant upregulation of *fixN*$_c$ expression (1.7-fold upregulated, p-value 0.101) in 21 day old bacteroids compared to free-living Rlv3841 [93]. This contrasts sharply with *fixN*$_9$ (38.1-fold up, p = 0.010) and *fixN*$_{10}$ (119.6-fold up, p = 0.003), both highly upregulated. Taken together, these findings suggest the two plasmid-encoded *fixNOQP* operons of Rlv3841 are functional and anaerobox-regulated, whilst *fixNOQP*$_c$ is not. Rlv3841 also has two homologs of the *fixGHIS* operon (>90% protein identity), encoding the assembly machine for the *fixNOQP* terminal oxidase [51,100]. Like *fixNOQP*, *fixGHIS* operons are typically anaerobox regulated [47,62,101]. Both *fixGHIS* operons in Rlv3841 have a single upstream anaerobox, in a near-identical position to those of the *fixNOQP* operons [85] (Table 1). In line with findings in other rhizobia, the *fixGHIS* and *fixNOQP* operons are therefore likely regulated by oxygen in a similar way [89,102,103].

Both *fixNOQP* operons were induced in cultured cells under microaerobic conditions, confirming their regulation by $O_2$ (Fig 2). In culture, the double *hfixL* mutant severely reduced this microaerobic induction, resulting in minimal expression of *fixNOQP*$_{10}$ and 17% of WT *fixNOQP*$_9$ expression (Fig 3). The single *hfixL*$_9$ mutant significantly reduced expression of both operons whilst the *hfixL*$_c$ mutant only reduced expression of *fixNOQP*$_9$, to 71% of WT. These results indicate hFixL$_9$ is the dominant protein and hFixL$_c$ plays only a minor role in *fixNOQP* expression, matching their respective importance for microaerobic *fnrN* expression (Fig 3). In the Rlv3841 *fxkR*$_9$ mutant, expression of both *fixNOQP* operons was reduced to less than 25% of WT, indicating that the protein is required for hFixL regulation of *fixK* and hence *fixNOQP*, as found in other rhizobia [44]. The hFixL-FxkR-FixK pathway is thus a key regulator of *fixNOQP* expression under free-living microaerobic conditions. The remaining expression of *fixNOQP*$_9$ and *fixNOQP*$_{10}$ in the *fxkR*$_9$ mutant may be due to redundancy via the *fxkR*$_c$ homolog, or the result of background FixK or FnrN activity. By contrast, in these conditions the *fnrN* mutant minimally affected *fixNOQP* expression, with only *fixNOQP*$_9$ showing a small albeit statistically significant reduction (73% of WT). In line with our study of *fnrN* expression, the hFixL-FxkR-FixK pathway is crucial for *fixNOQP* expression under free-living microaerobic conditions whilst FnrN plays a minimal role. It is likely that the FnrN protein remains mostly inactive at the $O_2$ concentration (1% $O_2$) used in our free-living experiments.

We also checked the activity of NifA, a central activator of nitrogen fixation genes, in free-living microaerobic conditions [64,65,104]. NifA is $O_2$ sensitive and in most rhizobia is active only in the near-anoxic core of nodules [32,55,105,106]. Work in *E. meliloti* has however suggested the protein may already be active in the early stages of nodule development [107]. Recent work has suggested some rhizobial *nifA* variants can be active outside of the nodule [108,109]. We checked the NifA dependant induction of *nifH*, a component of the nitrogenase complex [70,73,110]. As expected, *nifH* expression did not increase under microaerobic conditions (Fig 2), indicating Rlv3841 NifA is not expressed or is inactive under these conditions.

Next, the role of FnrN and hFixL on expression of *fixNOQP in planta* during symbiosis was studied. We found that nodules formed by the *fnrN* mutant expressed both *fixNOQP* operons at only 5% of WT (Fig 4B and 4C). The FnrN sensor is thus critical for *fixNOQP* expression

inside the nodule. In nodules infected by the double *hfixL* mutant, expression of *fixNOQP*$_9$ and *fixNOQP*$_{10}$ was reduced to 68% and 58% of WT, respectively. Expression of *fixK*$_{9a}$ was abolished (S1 Fig), suggesting minimal FixK production in the absence of hFixL-FxkR TCS activity. Taken together, our results indicate that FnrN is critical for *fixNOQP* expression during symbiosis but the hFixL-FxkR-FixK pathway also plays a significant role.

To assess the impact of FnrN and the hFixL-FxkR-FixK pathway on symbiotic nitrogen fixation, acetylene reduction assays were performed on pea plants inoculated with $O_2$ regulation mutants. In line with its poor expression of *fixNOQP*, the *fnrN* mutant was critically impaired in nitrogen fixation, reducing acetylene at only 15% of the WT level (Fig 5A). Plants inoculated with this mutant produced only small and unelongated pale or brown nodules indicative of poor development and low leghaemoglobin production (Fig 5D). Thus, FnrN is critical for effective nitrogen fixation by Rlv3841. Complementation restored 88% of WT acetylene reduction activity and produced nodules indistinguishable from WT (S2 Fig).

Plants inoculated with either individual *hfixL* mutants or the double mutant were also impaired in nitrogen fixation but retained approximately 75% of WT acetylene reduction activity (Fig 5A). No morphological changes were observed in these nodules (Fig 5C). Thus, the hFixL-FxkR-FixK pathway is also an important contributor to symbiotic fixation activity and is required to attain a WT level of fixation. Complementation of the double *hfixL* mutant was attempted but the gene was found to be toxic in *E. coli* (see Materials and Methods for details). The *fxkR*$_9$ mutant impaired acetylene reduction rates but the decrease was insufficient to be significant (p = 0.0584). The FxkR$_c$ homolog is likely at least partially active and sufficiently produced to rescue hFixL regulation in the absence of FxkR$_9$. In the triple *fnrN hfixL*$_9$ *hfixL*$_c$ mutant (LMB673; *fnrN*::ΩTet *hfixL*$_9$::ΩSpec *hfixL*$_c$:pK19) only negligible levels of fixation were recorded. This reinforces the importance of the contribution from both the hFixL-FxkR-FixK pathway and FnrN, suggesting no additional regulators exist which induce these anaerobox controlled genes in Rlv3841 during symbiosis.

## hFixL and FnrN are active in spatially distinct nodule zones during symbiosis

Legume nodules create a large internal $O_2$ gradient, with semi-aerobic conditions at their tip and near-anoxic conditions as low as 20 nM $O_2$ in the central nitrogen fixing zone [26,111]. This gradient is typically split into four zones (Fig 6A) containing different $O_2$ concentrations and rhizobia in different stages of differentiation (for reviews, see [26,31,112]). To understand how hFixL-FxkR-FixK and FnrN operate in this context, we used confocal microscopy to map the spatial expression of *fnrN* and *fixNOQP* in nodules.

Expression of *fnrN* in nodules infected with WT Rlv3841 (Fig 6B) was visible throughout all nodule zones. This included expression in infection threads in zone I, indicating that low $O_2$ induction of *fnrN* begins when Rlv3841 first enters the nodule and before the bacteria have differentiated into bacteroids. By contrast, *fnrN* expression in zone I was greatly reduced in nodules infected with the double *hfixL* mutant (Fig 6B). This suggests the $O_2$ concentration in the relatively aerobic environment of zone I is sufficiently low to activate the hFixL-FxkR-FixK pathway. In the absence of this pathway, some *fnrN* expression was retained in zone II and interzone II-III, but this was weaker than WT. Minimal *fnrN* expression was observed in zone III in the *hfixL* double mutant.

In the *fnrN* mutant, expression of *fnrN* appeared to be localized primarily in infection threads, around the entire periphery of the nodule (Fig 6B). Nodules infected by this mutant were severely impaired in their development, failed to elongate and contained little to no leghaemoglobin (Fig 5D). Free $O_2$ concentration is unlikely to drop as much in these nodules

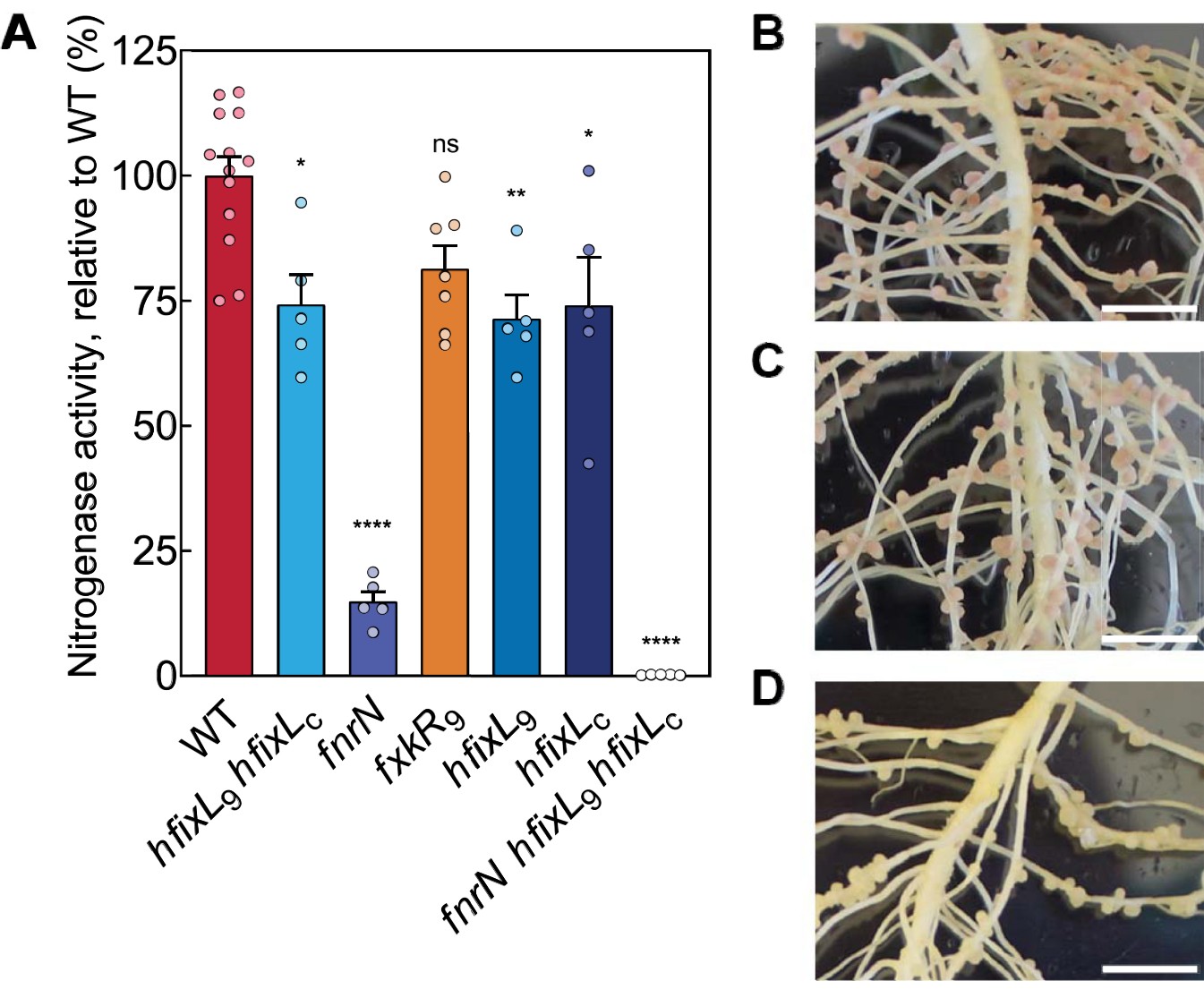

**Fig 5. Effect of oxygen regulation mutants on nodule morphology and acetylene reduction rates. (A)** Acetylene reduction rates of Rlv3841 mutant strains, normalised by WT activity (5.8 µmoles ethylene plant$^{-1}$ hr$^{-1}$, 16.8×10$^{-3}$ µmoles ethylene mg$^{-1}$ of nodules hr$^{-1}$). Knocking out the *hfixL* genes individually (*hfixL$_9$*, LMB495; *hfixL$_c$*, LMB403) and in combination (*hfixL$_9$ hfixL$_c$*, LMB496) only slightly reduced fixation. The *fnrN* mutant (LMB648) critically reduced fixation. The single *fxkR$_9$* mutant (OPS1808) did not significantly reduce fixation (p = 0.0584), possibly because of redundancy through the *fxkR$_c$* homolog. The mutant lacking both FnrN and hFixL-FxkR-FixK function (LMB673) fixed at only a negligible rate. Rates are normalised per plant to total mass of nodules. Data are averages (±SEM) from at least five plants, ns (not significant) P $\geq$ 0.05; *P < 0.05; **P < 0.01, ****P < 0.0001 by one-way ANOVA with Dunnett's post-hoc test for multiple comparisons. Photos of nodules colonized by WT **(B)**, the double *hfixL* knockout **(C)** and the *fnrN* knockout **(D)**. Scale bar, 1 cm.

as it does in fully developed nodules. It is therefore noteworthy that the hFixL-FxkR-FixK pathway is nevertheless active, suggesting even poorly developed nodules produce a sufficiently low O$_2$ concentration to activate the pathway.

Expression patterns of *fixNOQP$_9$* (Fig 7A) and *fixNOQP$_{10}$* (Fig 7B) were similar. In WT Rlv3841, expression of both started abruptly in the II-III interzone of nodules, in agreement with past studies [55,56,113,114]. This abrupt start was absent in nodules infected with the double *hfixL* mutant, indicating it requires the hFixL-FxkR-FixK pathway (Fig 7C and 7D). Without hFixL-FxkR-FixK, expression of *fixNOQP$_9$* and *fixNOQP$_{10}$* started gradually after the II-III interzone, presumably driven by FnrN. Expression was also weaker than in the WT. In

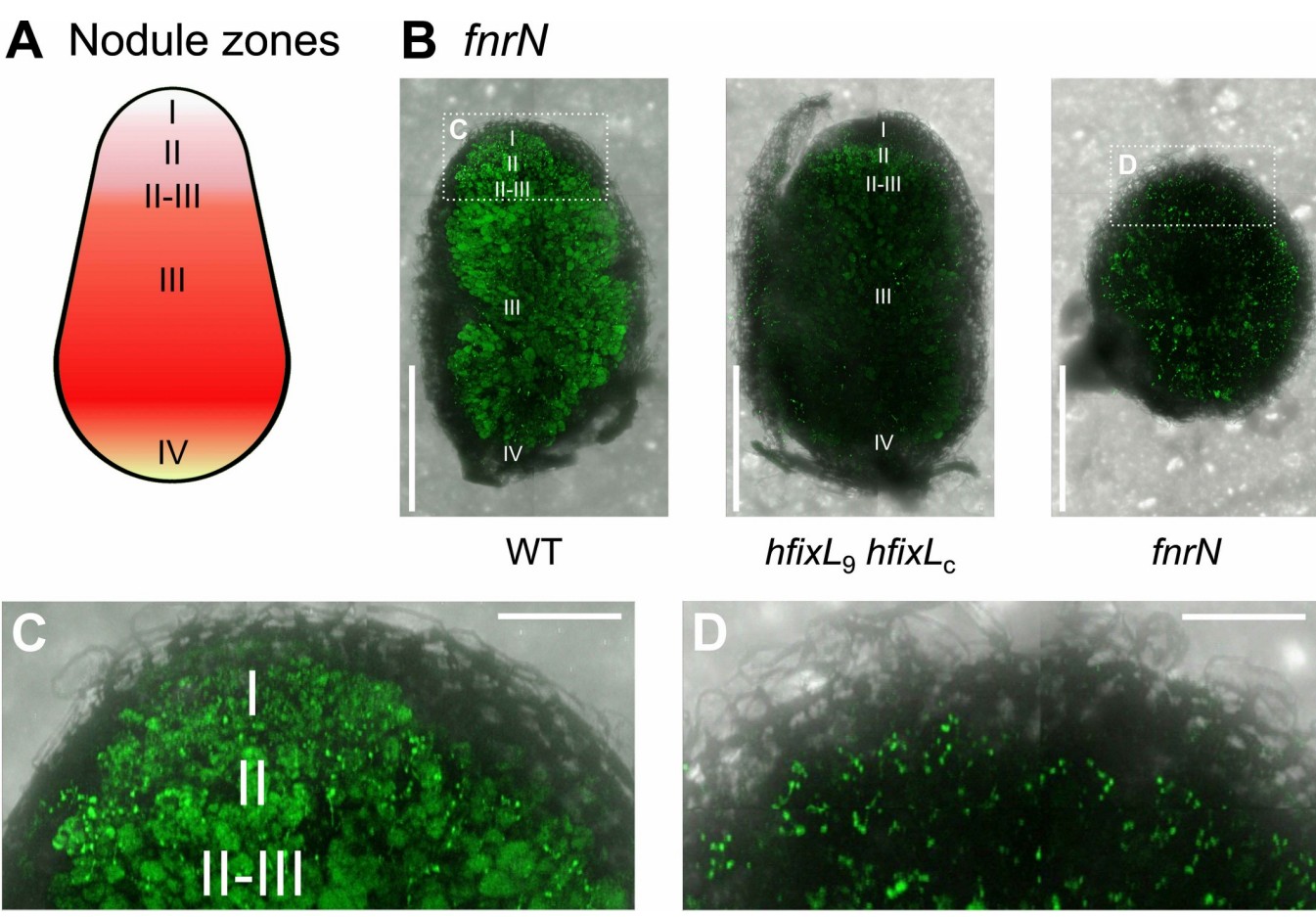

**Fig 6. Spatial expression pattern of *fnrN* in nodules infected with Rlv3841 WT and mutants.** (**A**) Schematic representation of an indeterminate nodule formed by *P. sativum*. Zone I contains undifferentiated rhizobia in infection threads. Rhizobia enter plant cells in zone II and in the II-III interzone undergo substantial differentiation towards becoming bacteroids. Zone III is the main nitrogen fixing zone. Zone IV contains bacteroids which are beginning to senesce. (**B**) Nodule cross sections showing expression of *fnrN* when inoculated with strains of Rlv3841 (Tn7 integrated *syfp2* promoter fusion: WT, OPS2429; *hfixL₉ hfixLc* double mutant, OPS2435; *fnrN* mutant, OPS2432). Expression begins immediately in zone I in nodules inoculated with WT; see C for a close-up of the region highlighted in white. A similar level of expression is present across all zones. When inoculated with the double *hfixL* mutant, expression began in zone II and was highest in this zone. In nodules inoculated with the *fnrN* mutant, expression was observed in infection threads around the periphery of the nodule; see D for a close up of the region highlighted in white. This mutant does not form mature nodules, and the normal zones are therefore unlikely to be fully developed. (**C**) Magnified view of *fnrN* expression in WT bacteria in zone I of the nodule. (**D**) Magnified view of *fnrN* expression in the infection threads of a nodule inoculated with the *fnrN* mutant. Scale bar; 1 mm (B), 0.25 mm (C and D). All images were captured and processed using identical parameters; see Materials and Methods for details.

the *fnrN* mutant, we observed minimal expression of *fixNOQP₉*. This may be due in part to the poor development of these nodules, but the expression of *fnrN* in the *fnrN* mutant (Fig 6B) indicates that the hFixL-FxkR-FixK pathway is still relatively active in these underdeveloped nodules. The lack of *fixNOQP* expression in the *fnrN* mutant therefore indicates that hFixL-FxkR-FixK cannot directly induce much *fixNOQP* expression in zone III of mature nodules. Although FnrN is the main driver of *fixNOQP* expression, the hFixL-FxkR-FixK pathway is required for full *fnrN* expression and also plays an important role, albeit indirectly.

### Integration of FnrN and hFixL improves the O₂ response

To study the dynamics of an integrated cascade containing both hFixL and FnrN, we constructed an ordinary differential equation model of their combined pathway based on past

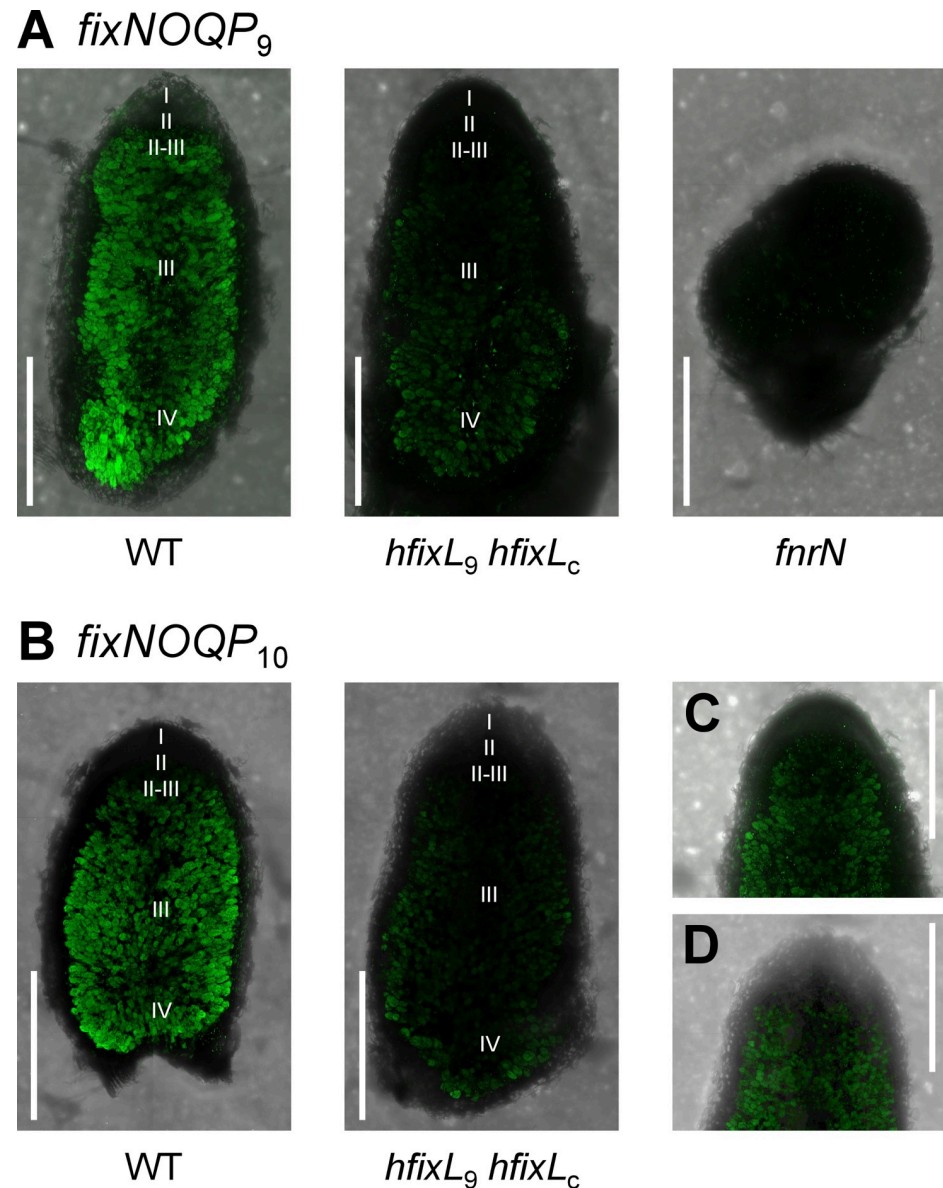

**Fig 7. Spatial expression pattern of the *fixNOQP* operons in nodules infected with Rlv3841 WT and mutants.** (**A**) Expression of *fixNOQP*$_9$ in strains of Rlv3841 (Tn7 integrated *syfp2* promoter fusion: WT, OPS2428; *hfixL*$_9$ *hfixL*$_c$ double mutant, OPS2434; *fnrN* mutant, OPS2431). In nodules inoculated with WT, expression starts abruptly at the II-III interzone. In the double *hfixL* mutant, expression is reduced and begins gradually and at a point more proximal to the root. Almost no expression is found in *fnrN* mutant nodules. (**B**) Expression of *fixNOQP*$_{10}$ in WT Rlv3841 and the double *hfixL* mutant followed a similar pattern as *fixNOQP*$_9$ (pJP2 reporter plasmid *syfp2* promoter fusion: WT, OPS2468; *hfixL*$_9$ *hfixL*$_c$, OPS2469). Expression begins at the II-III interzone. In the double *hfixL* mutant, expression again begins at a point more proximal to the root, in zone III of the nodule, and is reduced. (**C**) and (**D**) are areas of the *hfixL* double mutant reporter images (for *fixNOQP*$_9$ and *fixNOQP*$_{10}$ respectively) with their brightness and contrast altered to better display the distribution of fluorescence. Images within a set were captured and processed using identical parameters. Fluorescence intensity between the A and B image sets, and across C and D, should not be compared as intensity was normalised.

literature (S1 Text). A map of the regulatory connections incorporated in the model is given in S3 Fig. One each of hFixL, FxkR, FixK, FnrN and FixNOQP is considered in the model. hFixL was assumed to become active near a headspace concentration of 1% $O_2$ and FnrN near 0.01%

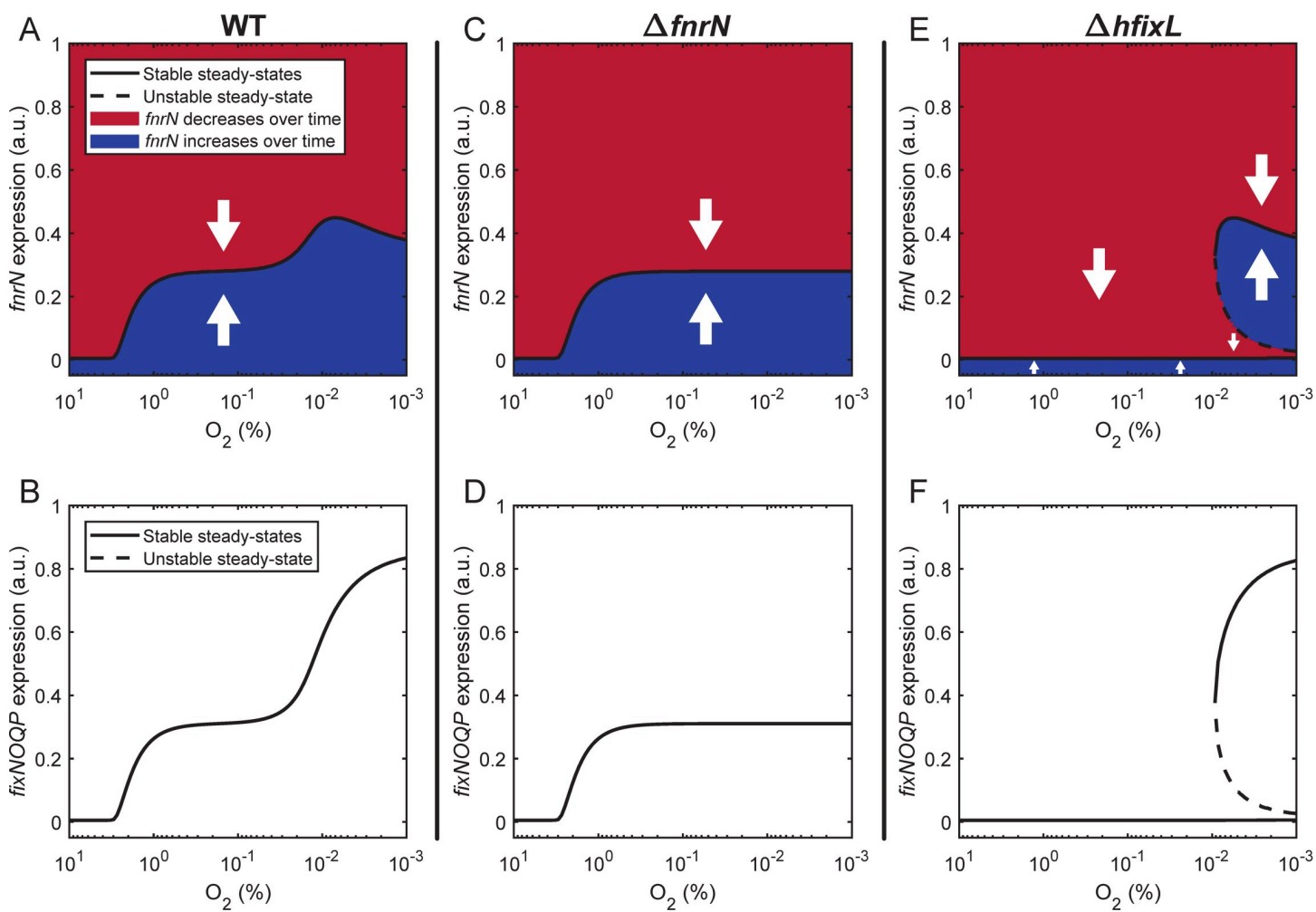

**Fig 8. Modelling predicts the biphasic response of *fnrN* and *fixNOQP* expression controlled by a cascade integrating the hFixL and FnrN oxygen sensors.** (A, C, E) Expression of *fnrN*. (**B, D, F**) Expression of *fixNOQP*. Black lines indicate stable expression steady-states, dashed lines indicate unstable steady states. In cells whose state is in a red shaded area, expression is expected to decrease; in a blue shaded area, it is expected to increase (white arrows indicate expected direction of change). (**A**) In a WT system with both sensors, expression of *fnrN* initially begins under microaerobic conditions. Expression increases slightly due to FnrN auto-activation as $O_2$ concentration drops, but is stabilized by auto-repression. (**B**) Expression of *fixNOQP* begins under microaerobic conditions, then increases when FnrN becomes active as $O_2$ drops further. (**C, D**) In the absence of FnrN, expression of *fnrN* and *fixNOQP* begins under microaerobic conditions driven by the hFixL-FxkR-FixK pathway, but does not subsequently increase. (**E, F**) In cells where only the FnrN system is present, expression of the *fnrN* gene and *fixNOQP* operon does not occur until $O_2$ concentration drops sufficiently for FnrN to be active. Once FnrN is active, the system exhibits bistability, with stable states of both near-zero and high levels of *fnrN* and *fixNOQP* expression. As the $O_2$ concentration continues to drop, an increasing proportion of cells are expected to transition to the high expression state due to stochastic variations in expression.

$O_2$, corresponding to dissolved $O_2$ concentrations of ~12 μM and ~120 nM respectively at equilibrium. Past studies have suggested that hFixL binds $O_2$ cooperatively, and that FnrN and FixK bind DNA as dimers [87,115–118]. Consequently all three of these binding processes were modelled using Hill functions with a Hill coefficient of 2 [119,120]. FixK and FnrN were assumed to have an identical induction effect on transcription when bound to the anaerobox motif. However, based on the critical role that FnrN but not FixK plays in regulation, FnrN was assumed to have a greater binding affinity for anaerobox motifs than FixK.

Our model reproduced the biphasic response of the integrated hFixL-FnrN cascade observed in WT Rlv3841 as $O_2$ dropped from atmospheric (~21%) to near-anoxic concentrations (0.001%) (Fig 8A and 8B). Expression of *fnrN* and *fixNOQP* first began in microaerobic

conditions (10–1% $O_2$) under the action of the hFixL-FxkR-FixK pathway. Subsequent activation of FnrN around 0.01% $O_2$ led to a further increase in *fixNOQP* expression, mirroring our findings *in planta*. Expression of *fnrN* near 0.01% $O_2$ initially increased due to auto-activation. Subsequently, as the $O_2$ concentration continued to drop, *fnrN* expression decreased due to auto-repression. Thus, the model correctly predicts a generally homogenous level of *fnrN* expression throughout the nodule, and increased *fixNOQP* expression in the core of the nodule relative to the tip.

In the absence of FnrN (Fig 8C and 8D), the model shows initial expression of *fnrN* and *fixNOQP* under microaerobic conditions due to the hFixL-FxkR-FixK pathway. There is however no further induction as $O_2$ continues to drop. This agrees with our finding that some *fnrN* expression takes place in the Rlv3841 *fnrN* mutant, albeit at a reduced level relative to WT. The model also predicts a lower level of *fixNOQP* expression in the *fnrN* mutant, consistent with our confocal microscopy results for *fixNOQP*9 expression (Fig 7A).

In the absence of *hfixL* (Fig 8E and 8F), an important new behaviour of the pathway is predicted by our model. As expected, no induction of *fnrN* or *fixNOQP* takes place under microaerobic conditions, in line with our experimental findings. However, once $O_2$ drops below 0.01%, our model suggests that the pathway may be bistable, with possible steady states at either high *fnrN* expression or near-zero expression. As the $O_2$ concentration continues to decrease, the disturbance needed to move from minimal expression to the high expression steady state becomes smaller. Thus, as the bacteria experience increasingly anaerobic conditions moving to areas of the nodule more proximal to the root, the model predicts that an increasing proportion of cells will transition from near-zero to high *fnrN* expression due to stochastic variations in expression of the gene. This agrees with the gradual increase in *fixNOQP* expression observed in the *hfixL* double mutant in Rlv3841 (Fig 7A and 7B).

## Discussion

$O_2$ regulation is essential for rhizobia to establish a successful symbiosis with their legume partners. The model *Rhizobium* Rlv3841 employs three $O_2$ sensors in symbiosis: hFixL, FnrN and NifA. In the present study, we examined this multi-sensor arrangement through a combination of *in vitro*, *in planta* and *in silico* approaches. The hFixL-FxkR-FixK pathway is active in the earliest stages of symbiosis, followed by FnrN as the bacteria move to areas of the nodule more proximal to the root. Both regulate genes required for symbiotic survival, such as *fixNOQP*. NifA is active at a later stage, in zone III of nodules, and regulates activation of core nitrogen fixation machinery. The hFixL-FxkR-FixK pathway is the most $O_2$ tolerant of the three sensors, active in free-living bacteria under microaerobic conditions and *in planta* beginning in zone I of nodules. The FnrN protein is inactive under free-living microaerobic conditions and only becomes active from the II-III interzone onwards. FnrN is critical for expression of *fixNOQP* and nitrogen fixation activity. Indirectly, the hFixL-FxkR-FixK pathway also plays an important role by inducing *fnrN* expression under microaerobic conditions, priming it for auto-activation in the central nitrogen fixing zone. Our modelling results suggest the hFixL-FxkR-FixK pathway also prevents bistability in the low $O_2$ response, thereby ensuring all cells commit to *fixNOQP* expression in the central nitrogen fixing zone. Thus hFixL-FxkR-FixK and FnrN act as a single regulation pathway which integrates both $O_2$ sensors.

Rhizobia experience a drop in $O_2$ concentration of at least three orders of magnitude as they transition from a free-living lifestyle in soil to terminally differentiated bacteroids in nodules. Like Rlv3841, it is common for other rhizobia to employ multiple $O_2$ sensors during this transition. These multiple sensors may be used to create redundancy, a feature often found in

key regulatory pathways to improve their robustness. Elements of this redundancy are present in Rlv3841, including the multiple hFixL homologs and the overlap between their role and that of FnrN. However, our results also demonstrate that each sensor plays an important distinct role. Thus, integrating sensors into a single cascade in Rlv3841 also improves the responsiveness of regulation and allows the bacteria to respond appropriately across the entire range of $O_2$ concentrations experienced during symbiosis. A similar dual-sensor arrangement has also previously been described in *Rhodopseudomonas palustris*, which combines a FixLJ-FixK pathway with the FnrN homolog AadR [121–123]. *R. palustris* is not symbiotic but is noted for its ability to grow under both aerobic and anaerobic conditions [124]. The combined pathway in *R. palustris* was shown to provide fine-tuned regulation for adapting to the large range of $O_2$ concentrations it experiences.

The prevalence of multi-sensor $O_2$ regulation arrangements in rhizobia may also have arisen in response to competitive fitness pressures. Legume plants can sanction rhizobia based on their nitrogen fixation activity [125–127]. We speculate the bacteria may also be selected based on the speed with which they are able to adapt to life inside nodules and begin productively fixing nitrogen. This would create pressure for strains to rapidly demonstrate their effectiveness to their legume host. Past work has suggested that one of the benefits of FnrN compared to FixLJ-FixK is that it is more responsive to $O_2$ concentration, providing more flexible regulation [86,87,128]. By enabling more fine-tuned control, integrated multi-sensor $O_2$ regulatory pathways may speed up the symbiotic transition, providing a competitive advantage.

## Materials and methods

### Bacterial strains and growth conditions

*E. coli* strains were grown in liquid or solid LB medium[129] at 37˚C supplemented with appropriate antibiotics (μg mL$^{-1}$): ampicillin 100, kanamycin 20, spectinomycin 50 and gentamicin 10. Rlv3841 strains were grown at 28˚C in Tryptone-Yeast (TY) extract[130] or Universal Minimal Salts (UMS)[131] with glucose and ammonium chloride at 10 mM each. Antibiotics for Rlv3841 were used at the following concentrations (μg mL$^{-1}$): gentamicin 20, kanamycin 50, spectinomycin 100, streptomycin 500, tetracycline 2, neomycin 80 and nitrofurantoin 20. A list of the strains used is given in S2 Text.

### Cloning, colony PCRs and conjugations

All routine DNA analyses were done using standard protocols [129]. PCR reactions for cloning were carried out according to the manufacturer's instructions with Q5 High-Fidelity DNA Polymerase (New England Biolabs). Colony PCRs used OneTaq DNA Polymerase (NEB). Restriction enzymes (NEB) were used according to the manufacturer's instructions. Sanger sequencing was carried out by Eurofins Genomics. Assemblies using BD In-Fusion cloning (Takara Bio) were performed according to the manufacturer's instructions. Triparental conjugations, and transductions with bacteriophage RL38, were performed as previously described [132,133]. Tn7 integrations were performed according to the method described by Choi and colleagues [134,135]. A list of the plasmids and primers used is given in S2 Text.

### Mutant generation and complementation

**Rlv3841 *hfixL*$_c$ (RL1879) mutant, LMB403.** A 1 Kb internal fragment of *hfixL*$_c$ was PCR amplified from Rlv3841 with primers pr0988/0989, adding XbaI sites at the 5' and 3' ends. This fragment was cloned into pK19mob digested with XbaI, using BD In-Fusion cloning, to

produce plasmid pLMB441. Triparental filter conjugation of pLMB441 into WT Rlv3841 was then performed using kanamycin selection. Colonies were screened by colony PCR using primers pr0482 and pK19A, which bind upstream of *hfixL*$_c$ and inside the integrated pK19 backbone respectively. This gave mutant strain LMB403.

**Rlv3841 *hfixL*$_9$ (pRL90020) mutant, LMB495.** A 1 Kb region containing *hfixL*$_9$ was PCR amplified from Rlv3841 with primers pr1270/1271. This region was subcloned into pJET1.2/blunt to produce plasmid pLMB581. Plasmid pLMB581 was then digested with XbaI/XhoI and the *hfixL*$_9$ region cloned into pJQ200SK using BD In-Fusion, digested with the same enzymes, to give plasmid pLMB585. A spectinomycin resistance cassette was digested out of the pHP45ΩSpc plasmid with SmaI and cloned into pLMB585 at a unique StuI site blunted using the Klenow fragment to give plasmid pLMB590. Triparental filter conjugation of pLMB590 into WT Rlv3841 was then performed using spectinomycin selection. Colonies were screened by colony PCR using primers pr1272/1273. This gave mutant strain LMB495.

**Rlv3841 double *hfixL*$_c$ *hfixL*$_9$ mutant, LMB496.** An Rlv3841 mutant in both *hfixL* genes was generated by triparental filter conjugation of pLMB441 into strain LMB495, producing double mutant strain LMB496.

**Rlv3841 *fnrN* (RL2818) mutant, LMB648.** A 2.5 Kb region containing *fnrN* was PCR amplified from Rlv3841 with primers pr1381/1382. This fragment was digested with XbaI/XhoI and cloned using BD In-Fusion into pJQ200SK linearized with digestion by the same enzymes to make plasmid pLMB732. A tetracycline resistance cassette was then digested out of the pHP45ΩTet plasmid with EcoRI and cloned into pLMB732 at a unique MfeI site to give plasmid pLMB733. Triparental filter conjugation of pLMB733 into WT Rlv3841 was then performed using tetracycline selection. Colonies were screened by colony PCR using primers pr1432/1433. This gave mutant strain LMB648.

**Rlv3841 triple *hfixL*$_c$ *hfixL*$_9$ *fnrN* mutant, LMB673.** A triple Rlv3841 mutant, in both *hfixL* genes and the *fnrN* gene, was generated by transducing *fnrN*::ΩTet from LMB648 into LMB496 to produce strain LMB673.

**Rlv3841 *fxkR*$_9$ (pRL90026) mutant, OPS1808.** Two 1 Kb regions, one upstream and one downstream of *fxkR*$_9$, were PCR amplified from Rlv3841 with primer pairs oxp2874/2875 and oxp2876/2877 respectively. These were cloned with BD In-Fusion into pK19mobSacB digested with PstI and EcoRI to produce plasmid pOPS1199. Triparental filter conjugation of pOPS1199 into WT Rlv3841 was then performed using kanamycin selection. Colonies were screened by colony PCR using primers oxp3155 and pK19A. Colonies with correct integration were subsequently subjected to sucrose selection to remove plasmid pK19mobSacB as previously described [136]. Colonies were then screened for loss of kanamycin resistance and using colony PCR with primers oxp3155/3156 to isolate mutant strain OPS1808.

**Complemented Rlv3841 *fnrN* mutant (OPS2260).** The *fnrN* gene with its native promoter was amplified from Rlv3841 with primers oxp4115/4116 and cloned into BsaI-digested pOGG280 using BD In-Fusion. This plasmid was then genomically integrated with kanamycin selection and colonies screened with primers oxp2327/2328 and confirmed with sequencing. This produced strain OPS2260.

**Attempts at complementing the Rlv3841 *hfixL*$_9$ mutant.** Our *hfixL*$_9$ (LMB495) mutant showed the largest phenotypic effect out of the two single *hfixL* mutants, and we therefore attempted to complement this strain. We first attempted complementation via Tn7 integration using the pOGG280 backbone in which *hfixL*$_9$ was under P*lac* control. This construct assembled in *E. coli* but could not be successfully conjugated into Rlv3841. We theorized the protein was being produced to a toxic level in Rlv3841 due to poor LacI repression. We next attempted to rectify this problem by driving *hfixL*$_9$ from the native *fixK*$_9$ promoter and RBS instead (*fixK*$_9$ and *hfixL*$_9$ likely form an operon). However, this construct could not be transformed

into *E. coli*, suggesting P*fixK*$_{9a}$ was causing toxic levels of *hfixL*$_9$ production. Finally, we sought to strike a balance between these two approaches by using the P*lac* promoter but the native *hfixL*$_9$ RBS, in a pOGG250 backbone. This construct assembled in *E. coli* and could be conjugated into Rlv3841 but failed to complement the mutant. It is likely that this promoter-RBS combination avoided toxicity by reducing *hfixL*$_9$ production but produced insufficient protein to achieve complementation.

## Microaerobic induction measurements in cultured cells

Rlv3841 strains were first grown on TY slopes with appropriate antibiotics for three days. Cells were resuspended and washed three times by centrifugation at 5,000 RCF for 10 minutes. Washed cells were used to inoculate 10 mL liquid UMS cultures to $OD_{600}$ 0.01 and grown overnight without antibiotics. Cultures were then diluted to OD 0.1 in 400 μL UMS per well in a 24-well microtiter plate (4titude). A gas-permeable membrane (4titude) was applied to microtiter plates. Plates were then incubated in a FLUOstar Omega plate reader equipped with an Atmospheric Control Unit (both produced by BMG) to adjust $O_2$ concentration to 1% and $CO_2$ concentration to 0.1%. Readings were taken every 30 minutes and plates shaken at 700 rpm in double orbital mode between readings. Induction was measured at 18 hrs post-inoculation, when all cultures had reached stationary phase.

## Plate-based measurements

sYFP2 measurements were made on a BMG FLUOstar Omega plate reader using the bottom optic with a gain setting of 2,000 and orbital averaging enabled (53 readings, 6 mm radius). Measurements were filter-based with excitation at 520 nm and emission recorded at 540 nm. Luminescence measurements were made on a Promega GloMax plate reader using the manufacturer's protocol. Luminescence was used for S1 Fig as this reporter construct was already available.

## Plant growth and acetylene reduction

*Pisum sativum* cv. Avola seeds were surface sterilized using 95% ethanol and 2% sodium hypochlorite before sowing. Plants were inoculated with $1 \times 10^7$ cells of the appropriate rhizobial strain and grown in 1 L beakers filled with sterile medium-grade vermiculite and nitrogen-free nutrient solution as previously described in a growth room (16h light/8h dark) [137]. Harvesting was 21 days later and acetylene reduction rate was determined as previously described [138]. All nodules for each plant were counted and their combined mass weighed; acetylene reduction rates were normalised by total nodule weight.

## Bacteroid isolation

Bacteroids were isolated from root nodules after 21 days of plant growth following a differential spin protocol adapted from Tsukada et al. 2009 [139]. Approximately 100 mg of nodules were picked per plant. Nodules were immersed in 1 mL of sterile isolation buffer (1 M $K_2HPO_4$, 1 M $KH_2PO_4$, 300 mM sucrose, 2 mM $MgCl_2$) and macerated. The mixture was spun down at 200 RCF for 5 minutes to remove plant debris. The supernatant was transferred to a fresh tube and spun down at 3,500 RCF for 5 minutes. The supernatant from this second spin was discarded and the pelleted fraction, containing the isolated bacteroids, was resuspended in isolation buffer and used for microtiter plate measurements.

### Transcription start site mapping

The TSS data set referenced in this paper can be found in full on the NCBI SRA database, Bio-Project number PRJNA667846. A publication discussing the data in full is forthcoming. Protocol details can be found in S3 Text.

### Statistical analysis

All analyses were performed using GraphPad Prism 8 (GraphPad Software). Significant differences were determined by Student's t-test or one-way ANOVA followed by Dunnett's multiple comparisons post-hoc test correction. A p-value less than 0.05 was considered statistically significant.

### Confocal microscopy

Reporters were constructed by transcriptional fusion of promoters to an ORF of the sYFP2 fluorescent protein. Reporters were subsequently genomically integrated into Rlv3841 strains using the mini-Tn7 system [134]. Plants were inoculated with marked strains and grown as described above. After 21 days, nodules were picked and immersed in water then cut in half longitudinally. Images were taken with an LSM 880 confocal laser-scanning microscope equipped with the Axio Imager.Z2 (Zeiss), using the manufacturer's ZEN Black software. A Plan-Apochromat 10×/0.45 M27 objective (Zeiss) was used. Excitation was at 514 nm with an Argon laser and emission measurements filtered to a range of 519–572 nm. Acquisitions were tile scans with 2×3 tiles per image. 31 Z-stack slices were taken for each tile, separated by a height of 10 μm. Images shown in this publication are maximum-intensity orthogonal projections produced with the ZEN Blue software (Zeiss).

### Figure data

Data for main text Figs 2–5 and S1 and S2 Figs are given in S4 Text. Data for main text Fig 8 are given in S1 Spreadsheet.

## Supporting information

**S1 Fig. hFixL is required for *in planta fixK*9a expression in Rlv3841.** A reporter (pOPS0136) was built with the *luxCDABE* reporter operon fused to the *fixK*9a promoter. The promoter was active in isolated WT Rlv3841 bacteroids (OPS0376), but no luminescence above no-reporter background was recorded in double *hfixL* mutant bacteroids (OPS0528). Data are averages (±SEM) from at least four plants, *P < 0.05; by one-way ANOVA with Dunnett's post-hoc test for multiple comparisons.
(EPS)

**S2 Fig. Complementation of the Rlv3841 *fnrN* mutant. (A)** Acetylene reduction rates; the activity of the *fnrN* mutant (LMB648) was 20% of WT Rlv3841. The complemented strain (OPS2260) fixed at 88% of WT. Nodules colonized by Rlv3841 **(B)** WT, **(C)** the complemented *fnrN* mutant and **(D)** the *fnrN* mutant. Acetylene reduction rates are normalised to total weight of nodules per plant. Data are averages (±SEM) from seven plants, ns (not significant) P ≥ 0.05; ****P < 0.0001; by one-way ANOVA with Dunnett's post-hoc test for multiple comparisons. Complementation also restored nodule morphology.
(EPS)

**S3 Fig. Simplified map of the Rlv3841 dual sensor oxygen cascade used for modelling.** Only one copy each is included of *hfixL*, *fixK*, *fxkR* and *fixNOQP*. Both FnrN and FixK can

positively and negatively regulate expression of *fnrN*. FxkR negative auto-regulation is not included in the model. Regulation is indicated with lines ending in arrows (positive regulation) and ending in blunt ends (negative regulation). Translation is shown as lines ending in circles. (EPS)

**S1 Text. Modelling oxygen regulation in Rlv3841.**
(PDF)

**S2 Text. Strains, plasmids and primers used in the study.**
(PDF)

**S3 Text. Further materials and methods details for transcription start site mapping.**
(PDF)

**S4 Text. Data tables for main text Figs 2–5 and S1 and S2 Figs.**
(PDF)

**S1 Spreadsheet. Data table for main text Fig 8.**
(XLSX)

# Acknowledgments

The authors would like to thank Dr Tim Haskett, Dr Carmen Sánchez-Cañizares and Prof Lee Sweetlove for their advice and critically reviewing the manuscript. They would also like to thank Dr Niloufer Irani for her help with confocal microscopy, and Dr Beatriz Jorrín for her help with the Tn7-based reporters used in this study.

# Author Contributions

**Conceptualization:** Paul J. Rutten, Harrison Steel, Graham A. Hood, Antonis Papachristodoulou, Philip S. Poole.

**Data curation:** Paul J. Rutten, Harrison Steel, Graham A. Hood, Vinoy K. Ramachandran.

**Formal analysis:** Paul J. Rutten, Harrison Steel, Vinoy K. Ramachandran.

**Funding acquisition:** Antonis Papachristodoulou, Philip S. Poole.

**Investigation:** Paul J. Rutten, Graham A. Hood, Vinoy K. Ramachandran, Lucie McMurtry, Barney Geddes.

**Methodology:** Paul J. Rutten, Harrison Steel, Graham A. Hood, Vinoy K. Ramachandran, Antonis Papachristodoulou, Philip S. Poole.

**Project administration:** Paul J. Rutten, Antonis Papachristodoulou, Philip S. Poole.

**Resources:** Harrison Steel, Graham A. Hood, Barney Geddes, Antonis Papachristodoulou, Philip S. Poole.

**Software:** Harrison Steel, Antonis Papachristodoulou.

**Supervision:** Barney Geddes, Antonis Papachristodoulou, Philip S. Poole.

**Validation:** Paul J. Rutten, Vinoy K. Ramachandran, Lucie McMurtry, Barney Geddes.

**Visualization:** Paul J. Rutten, Harrison Steel.

**Writing – original draft:** Paul J. Rutten, Harrison Steel.

**Writing – review & editing:** Paul J. Rutten, Harrison Steel, Graham A. Hood, Vinoy K. Ramachandran, Lucie McMurtry, Barney Geddes, Antonis Papachristodoulou, Philip S. Poole.

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
