## [Decision Letter · Decision Letter 0]

21 Oct 2020

Dear Dr Poole,

Thank you very much for submitting your Research Article entitled 'A multi-sensor system provides spatiotemporal oxygen regulation of gene expression in a Rhizobium-legume symbiosis' to PLOS Genetics. Your manuscript was fully evaluated at the editorial level and by independent peer reviewers. The reviewers appreciated the attention to an important topic but identified some aspects of the manuscript that should be improved.

We therefore ask you to modify the manuscript according to the review recommendations before we can consider your manuscript for acceptance. Your revisions should address the specific points made by each reviewer.

[LINK]

Yours sincerely,

Sean Crosson

Associate Editor

PLOS Genetics

Lotte Søgaard-Andersen

Section Editor: Prokaryotic Genetics

PLOS Genetics

Reviewer's Responses to Questions

**Comments to the Authors:**

Reviewer #1: This paper examines the multiple regulators and pathways that respond to oxygen in Rhizobium leguminosarum strain 3841, which is now the most frequently used strain and was the first to have its genome sequenced. Previous work in this area used other strains and there are signficant strain to strain variations in the complement of these regulators. Previous analyses also did not examine all components simultaneously , and make all the single double and triple mutants required for this analysis. The results obtained here show that the hfixL mediated pathway responds to higher concentrations of oxygen and is important in microxic conditions such as would arise in soils or in early steps of symbiosis. Inside the nodule, as oxygen concentration decreases, fnrN becomes the major regulator, and, in terms of fixNOPQ (encode a high oxygen affinity electron transport chain) induction, play the major role during N fixing symbiosis. Very nice fluoresence microscopy supports these results and a mathematical model is developed to account for the date and make predictions.

Overall this was a very strong paper that finally clariifies a lot of the contradictions inherent in the literature in this field. I don't have a lot to find fault with here.

One criticism is that the role of key regulator NifA is not discussed enough. nifA does not seem to be induced by any of these systems as far as I can tell (whereas in S. meliloti it is regulated by the FixLJ pathway), and it is not clear whether nifA is constitutive and oxygen sensing, autoinducing (read-through from other target genes ?) , or regulated by something else.

A second criticism is that no mention is made of the fixGHIS operons (there are two, one on each of plasmids pRL10JI and pRL9JI) and how they are regulated. Presumably this is by the same pathway as fixNOQP but it would have been easy to verifiy this in the mutants by qRTPCR or making a fixGHIS fusion.

Finally the role of RpoN in this regulatory pathway is completely ignored. Previous work has shown that rpoN positively regulates the fnrN promoter, and thus indirectly fixNOQP and fixGHIS (Clark et al. 2001, Molecular and General Genetics 264:623.

Other minor issues.

Line 119. Should point out to someone not familiar with the 3841 genome and gene numbering that one of these is a chromosomal copy, whereas the other is on pRL9JI, the third smallest plasmid, which also has complete copies of fixNOQP and fixGHIS operons.

Lines 119-123. How similar to each other are these alleged copies of the genes ? I have seen it suggested, for instance that R. leguminosarum has 3 copies of fixNOQP (also mentioned in this paper), but the chromosomal one bears relatively little similarity to the two on plasmids.

Fig 2. IS the fluorescence here from YFP ? Is there no decline in fluorescence due to low O2 that would affect these results ? legend should state that gene fusions are being used here, if that was indeed the case. It is clear that these are yfp fusions if one reads the supplementary table, but I don't think it should be nececssary to do that.

Line 370 LB does not actually (or did not originally) stand for Luria Bertani (it was Lysogeny broth - as pointed out by those authors themselves) but as the recipe has changed is best just left as LB with a reference.

Reviewer #2: Please see my review uploaded as an attachment

Reviewer #3: The manuscript by Rutten et al is on the whole a very well written manuscript that was nice to read. I very much commend the authors for a very concise and thorough introduction. This was really appreciated. The modeling of the response together with the genetics was a novel approach that I really appreciated.

My major criticism is the presentation of the data. In all honesty I am taking the conclusions on faith. What I found very difficult is that although the alleles were clearly marked on the figures, I found it impossible to figure out which strain was used for the actual experiment. I took a good deal of time going through the strain list. It was not easy to follow. This comment can be used for almost every figure that presented assay results. I assumed that the fluorescence measurements were made from a reporter that was chromosomally integrated. However that is completely an assumption.

Major concerns

Figure legends need to be rewritten so that the reader can interpret the data. At the very least if the strains numbers were included it would be able to follow the experiment.

Figure 4-are those strains really deletion strains that are marked with a delta symbol? My reading suggested they were insertional mutants. Although they may give identical results, they are not genetically equivalent.

Figure 5-It would be easier to interpret if the strains were mentioned in the figure legend.

Line 236-What is the evidence for toxicity? Was it an issue of how it was constructed? I really think something more should be given here. Is this an issue of no colonies were obtained on transformation? It is the feeling I am left with.

References-check for uniformity with respect to how the literature is cited -titles with capital letters or written in sentence form. Also check for italicization of bacterial names.

Reference 11 (line 501) seems incomplete. I would not know how to look up this work based on what is presented

Strain list can also be made more reader friendly. Inclusion of more description in the relevant characteristics into strains that carried plasmids would go a long way. I do admit that the plasmids themselves are well described in this section.

**Have all data underlying the figures and results presented in the manuscript been provided?**

Reviewer #1: Yes

Reviewer #2: **No: **It appears that the numerical data for the graphs were not provided in spreadsheet form.

Reviewer #3: Yes

PLOS authors have the option to publish the peer review history of their article (what does this mean?). If published, this will include your full peer review and any attached files.

Reviewer #1: No

Reviewer #2: No

Reviewer #3: No

---

## [Decision Letter · Decision Letter 1]

4 Dec 2020

Dear Dr Poole,

We are pleased to inform you that your manuscript entitled "Multiple sensors provide spatiotemporal oxygen regulation of gene expression in a Rhizobium-legume symbiosis" has been editorially accepted for publication in PLOS Genetics. Congratulations!

Yours sincerely,

Sean Crosson

Associate Editor

PLOS Genetics

Lotte Søgaard-Andersen

Section Editor: Prokaryotic Genetics

PLOS Genetics

Comments from the reviewers (if applicable):

Reviewer's Responses to Questions

**Comments to the Authors:**

Reviewer #1: The authors have made every effort to address the concerns of the reviewers, and have certainly addressed my concerns to my satisfaction. While it would have been useful to have had access to a marked up version of the paper where the changes were more obvious, I did compare the two versions, and found the changes to figure legends and other material to be a big improvement.

Reviewer #2: This is a review of a revised, resubmitted manuscript. I was one of the original reviewers and have critically read the reviewers' comments, authors' responses to those comments, and the new manuscript. This revised manuscript is excellent, and I applaud the authors for their thorough and careful revising. The TSS data table, additions to figures, inclusion of strain and plasmid names, new references, and rewriting have substantially improved what was already a nice manuscript. I also appreciated the authors' clear and detailed responses to the reviews, which made my task easier.

I have only a few minor comments:

1. Figure S1 has some extraneous faint gray lines to the right and bottom of the chart. Similarly, Figure S3 has faint gray lines and also faint gray text partially obscured by "PfnrN", "PfixNOQP1" and "PfxkR".

2. Lines 285-286 – for clarity, I suggest adding an "is", e.g., "NifA is not expressed or is inactive"

3. Page 7 of Supplementary 2 has two instances of "in vivo" which could be amended as the authors described in their response to reviewers.

4. On the review webform, I was asked to answer, "Have all data underlying the figures and results presented in the manuscript been provided?" and to confirm that "numerical data that underlies graphs or summary statistics" were provided.

The data tables for Figures 3 and 5 (Supplementary 4) give values as "% of the WT average", I was a bit unsure if these % values should be considered "numerical data" or "summary statistics". I lean toward the former, so I answered affirmatively on the form but maybe the editor wishes to weigh in on this.

Reviewer #3: I commend the authors for the thoroughness of this revision. The effort into improving the figure legends as well as the strain list is greatly appreciated and make manuscript much easier to interpret.

**Have all data underlying the figures and results presented in the manuscript been provided?**

Reviewer #1: Yes

Reviewer #2: Yes

Reviewer #3: Yes

PLOS authors have the option to publish the peer review history of their article (what does this mean?). If published, this will include your full peer review and any attached files.

Reviewer #1: No

Reviewer #2: No

Reviewer #3: No

**Data Deposition**

http://datadryad.org/submit?journalID=pgenetics&manu=PGENETICS-D-20-01363R1

**Press Queries**

---

## [Editor Report · Acceptance letter]

29 Jan 2021

PGENETICS-D-20-01363R1 

Multiple sensors provide spatiotemporal oxygen regulation of gene expression in a *Rhizobium*-legume symbiosis 

Dear Dr Poole, 

We are pleased to inform you that your manuscript entitled "Multiple sensors provide spatiotemporal oxygen regulation of gene expression in a *Rhizobium*-legume symbiosis" has been formally accepted for publication in PLOS Genetics! Your manuscript is now with our production department and you will be notified of the publication date in due course.

With kind regards,

Alice Ellingham

PLOS Genetics

On behalf of:
